# Antibacterial and Photocatalytic Activity of ZnO/Au and ZnO/Ag Nanocomposites

**DOI:** 10.3390/ijms242316939

**Published:** 2023-11-29

**Authors:** Mariana Busila, Viorica Musat, Petrica Alexandru, Cosmin Romanitan, Oana Brincoveanu, Vasilica Tucureanu, Iuliana Mihalache, Alina-Viorica Iancu, Violeta Dediu

**Affiliations:** 1Centre of Nanostructures and Functional Materials-CNMF, Faculty of Engineering, “Dunarea de Jos” University of Galati, Romania, Domneasca Street 111, 800201 Galati, Romaniapetrica.alexandru@ugal.ro (P.A.); 2National Research and Development Institute in Microtechnologies–IMT Bucharest, 126A Erou Iancu Nicolae Street, 077190 Bucharest, Romania; cosmin.romanitan@imt.ro (C.R.); oana.brincoveanu@imt.ro (O.B.); iuliana.mihalache@imt.ro (I.M.); 3Department of Morphological and Functional Sciences, Faculty of Medicine and Pharmacy, “Dunarea de Jos” University, 800008 Galati, Romania; 4Medical Laboratory Department, Clinical Hospital for Infectious Diseases “Sf. Cuvioasa Parascheva”, 800179 Galati, Romania

**Keywords:** ZnO/Au, ZnO/Ag, nanocomposites, antimicrobial, photocatalytic activity

## Abstract

The use of a combination of nanoparticles as antimicrobial agents can be one strategy to overcome the tendency of microbes to become resistant to antibiotic action. Also, the optimization of nano-photocatalysts to efficiently remove persistent pollutants from wastewater is a hot topic. In this study, two composites ZnO/Au (1% wt.) and ZnO/Ag (1% wt.) were synthesized by simple aqueous solution methods. The structure and morphology of the r nanocomposites were analyzed by structural and optical characterization methods. The formation of AuNPs and AgNPs in these experiments was also discussed. The antimicrobial properties of ZnO, ZnO/Au, and ZnO/Ag nanomaterials were investigated against Gram-negative bacteria (*Pseudomonas aeruginosa*) and Gram-positive bacteria (*Staphylococcus aureus*). The results showed an increase of 80% in the antimicrobial activity of ZnO/Au against *Pseudomonas aeruginosa* compared with 30% in the case of ZnO/Ag. Similarly, in the case of the *S. aureus* strain tests, ZnO/Au increased the antimicrobial activity by 55% and ZnO/Ag by 33%. The photocatalytic tests indicated an improvement in the photocatalytic degradation of methylene blue (MB) under UV irradiation using ZnO/Au and ZnO/Ag nanocomposites compared to bare ZnO. The photocatalytic degradation efficiency of ZnO after 60 min of UV irradiation was ∼83%, while the addition of AuNPs enhanced the degradation rate to ∼95% (ZP2), and AgNP presence enhanced the efficiency to ∼98%. The introduction of noble metallic nanoparticles into the ZnO matrix proved to be an effective strategy to increase their antimicrobial activity against *P. aeruginosa* and *S. aureus*, and their photocatalytic activity was evaluated through the degradation of MB dye. Comparing the enhancing effects of Au and Ag, it was found that ZnO/Au was a better antimicrobial agent while ZnO/Ag was a more effective photocatalyst under UV irradiation.

## 1. Introduction

In recent decades, engineered nanomaterials have been obtained and tested in new and different applications (medicine [1], environment [2], energy [3], etc.). Synthesis methods have been intensively developed so that a great variety of nanostructures can be obtained [4]. Nanoparticles (NPs) have specific physical–chemical characteristics like a high surface-to-volume ratio, high reactivity, and amenability to surface modification [5]. These properties recommend them for antimicrobial applications, being able to effectively target and kill Gram-positive and Gram-negative bacteria. Nowadays, the accelerated spread of antibiotic resistance and the small number of newly released antibiotics have caused a strong preoccupation to develop new strategies to defeat this public health threat [6]. NPs have already proven that they can be an alternative to classic antimicrobial agents, being unlikely to induce antimicrobial resistance [7,8], although some forms of resistance in the case of Ag have been reported [9,10]. Although there is extensive research, the antibacterial mechanism of nanostructured materials is not fully elucidated [11], and three plausible routes have been proposed: (i) reactive oxygen species (ROS) generation, (ii) the release of metal ions in aqueous media, and (iii) electrostatic interaction with the cell wall. 

Among antimicrobial nanomaterials, ZnO has been reported to exhibit broad-spectrum antibacterial properties against [12] viruses [13,14,15] and fungi [16,17], and is generally regarded as safe for humans. The interest in ZnO is facilitated by the relative ease of the synthesis of different kinds of morphologies [9]. Nano-ZnO can act against bacteria mainly through ROS generation, where the electron–hole pairs generated by the light irradiation create free radicals which, consequently, destroy cellular constituents. The wide band gap of ZnO results in a relatively low ROS production in the visible range, being more active in the UV range. One approach aiming to extend the light absorption to the visible region is coupling with plasmonic metals [18]. With the formation of the metal–semiconductor interface, the semiconductor’s ability to absorb light in the visible range is improved and the recombination of photogenerated charge carriers is hindered, leading to the augmentation of ROS generation. Different ZnO/noble metal nanocomposites have been increasingly developed, mainly for photocatalytic applications [19,20]. The usage of these plasmonic nanocomposites against microbes is a hot topic [18], and research still needs to be undertaken to yield results that can be transferred into practical applications. Au and Ag can enhance the photocatalytic antimicrobial action of ZnO, and they can also enhance treatment efficiency due to their intrinsic antimicrobial activity [21]. In a previous study, we demonstrated that ZnO/Au nanocomposites, composed of small ZnO nanorods and small AuNPs, work through a synergistic action, proving to be more effective against *E. coli* and *S. aureus* than ZnO alone [22]. Comparing the two metals, AuNPs have the advantage of a high degree of stability and biocompatibility, while AgNPs have shown undeniable antimicrobial qualities. Hernández-Sierra et al. reported a comparative investigation of the bactericidal activity of Ag, ZnO, and Au nanoparticles on *Streptococcus mutans* (*S. mutans*) [23] and showed that Ag nanoparticles exhibited the most activity in controlling *S. mutans*, the main cause of dental caries. AgNPs deposited onto ZnO proved to have better biocompatibility than metallic nanoparticles while preserving their antimicrobial properties [24,25]. 

Another promising application of nano-ZnO is in photocatalysis, as an eco-friendly and effective photoactive material that can degrade different recalcitrant organic compounds pollutants from wastewater [26]. Conventional wastewater treatment methods do not completely mineralize, even after a long processing time. The key aspect of photodegradation processes over ZnO nanostructures is the utilization of photoexcited charge carriers to break down selected contaminants into non-toxic substances. The photocatalytic removal of organic compounds implies the adsorption of the organic pollutants on the surface of the catalyst, oxidation, and reduction reactions at the interface, triggered under various light irradiations, resulting in the complete mineralization of the pollutant. ZnO has emerged as one of the most preferable photocatalyst candidates because of its strong UV absorption and high initial activity rate; also, it is safe for the environment and cheaper to produce than TiO_2_. However, the practical application of ZnO is hindered due to its large band gap energy, being active mostly under UV irradiation, and due to the rapid recombination of the photoinduced electron–hole pairs, faster than the surface redox reactions [27]. One promising approach to modulate the absorption of ZnO and to overcome the rapid recombination of charge carriers is the coupling of noble metals with nano-ZnO [28,29,30], using the surface plasmon resonance properties of noble metals [8]. Au and Ag nanoparticles can extend the light absorption to the visible range and can prolong the lifetime of photogenerated electrons and holes. The Schottky barrier established at the metal–semiconductor interface can increase photocatalytic activity by preventing the recombination of photogenerated electrons and holes, increasing the interfacial charge transfer between the noble metal and semiconductor, and prolonging the carrier lifetime [31,32]. Upon modification with Au or Ag, the metal centers act as an electron sink/trap in ZnO, increasing the photocatalysis efficiency under visible and UV irradiation. The weak electrostatic interactions between noble metals and ZnO are the main challenge for the synthesis of nanocomposites with a good dispersion of metal nanoparticles in the oxide matrix. It is important to ensure the heterogeneous growth of noble metal nanoparticles on ZnO to create metal–ZnO heterojunctions. It was demonstrated that a small quantity of noble metals is needed to induce improvements in organic degradation time. Adding 1% Ag to a ZnO photocatalyst-coated microreactor led to 91% degradation of methylene blue dye from water under UV-A irradiation for 2.5 min [33]. Other researchers found that the content of 1.5% AgNPs in nanocomposites was the most favorable for maximizing the photocatalytic performances, and that exceeding this percentage decreased the photodegradation efficiency [31]. Various Au/ZnO heterostructures with different gold contents were reported as potential materials for photocatalysis. In reference [34], a 1.30% Au content in ZnO/Au nanomaterial achieved the best performance in the degradation of methylene blue dye and was superior compared to the effect of silver on ZnO effectiveness. Other researchers found that silver had a greater positive influence than gold in the efficiency of the photocatalytic degradation of Congo red dye under UV irradiation [29]. When employed as photocatalysts for methylene blue degradation, it was found that Au (3%)/ZnO exhibited superior performance than the pure ZnO [35]. 

In this article, we propose two types of nanocomposites, ZnO/Au and ZnO/Ag, with 1 wt.% noble metal content, and we compare their antimicrobial and photocatalytic performance. We synthesized the nanocomposites through the reduction of Au or Ag salts using citric acid in the presence of citrate-functionalized ZnO nanoparticles. The antibacterial activity of these nanomaterials against *Pseudomonas aeruginosa* (*P. aeruginosa*) and *Staphylococcus aureus* (*S. aureus*) was investigated. The photocatalytic activity was measured by the degradation of methylene blue (MB) cationic dye in an aqueous solution. The objective of this study was to improve the antimicrobial and photocatalytic activity of ZnO through the introduction of reduced amounts of gold and silver into the ZnO matrix. Such a comparative study will eventually lead to the better utilization of these nanoparticles in killing dangerous bacteria or in the removal of toxic dyes from wastewater. 

## 2. Results and Discussion

The obtained nanomaterials received the codes according to Table 1 and were morphologically and functionally characterized by different techniques.

### 2.1. Characterization of the ZnO/Au and ZnO/Ag Nanocomposites

To assess the structural features of the investigated samples, X-ray diffraction investigations were performed. Figure 1 presents the experimental XRD patterns, as well as the simulated data in the framework of the Rietveld refinement [36]. The diffractograms of the sample correspond to the wurtzite structure of zinc oxide, and other diffraction patterns corresponding to impurities or sodium citrate were not found, suggesting the high purity of the synthesized nanomaterials.

This refinement is based on the least-squares method of the theoretical profile against the experimental XRD spectrum. Briefly, the principle of the refinement lies in the minimization of a function M, which accounts for the difference between a calculated profile $y_{i}^{calc}$ and the observed data $y_{i}^{obs},$ with the following form [37]:(1)$$M=\sum W_{i}{\left\{y_{i}^{obs}-\frac{1}{S}y_{i}^{calc}\right\}}^{2}$$
where $W_{i}$ is the statistical weight and S is an overall scale factor such that $y_{i}^{calc}=Sy_{i}^{obs}$. The values of the fitting parameters, $R_{wp}$ (weighted parameter) and S (scale factor), are presented in Figure 1, and their goodness of fit is proved by reasonable values, i.e., $R_{wp}$~10% and S~1. The values for the unit cell parameters, crystallite size, and lattice strain determined using the Rietveld fit are listed in Table 2. 

As can be seen from Table 1, the ZnO unit cell parameters were not affected during AuNP and AgNP synthesis, preserving the wurtzite crystal structure of ZnO with a = b = 0.32 nm, and c = 0.52 nm. The crystallite size of the commercial ZnO sample (unmodified) was 23.2 nm, and after citrate functionalization reached 31.1 nm. During the synthesis of AuNPs and AgNPs, the mean crystallite size of ZnO decreased to 27.5 nm in ZP2 and 25.3 nm in ZP3. This further implied an increase in the dislocation density (ρ), according to the following formula: ρ ~ 1τ2 , where τ is the mean crystallite size determined above [38]. At the same time, the lattice strain decreased from 0.73 to 0.42%, indicating a relaxation of the lattice simultaneously with the dislocation formation in the lattice. Overall, XRD analysis proved that the Au and Ag formation on ZnO NPs led only to a small worsening of the crystal quality by the formation of dislocations in the ZnO lattice. 

SEM micrographs of ZnO and composite samples are shown in Figure 2. As can be seen, ZnO particles appeared as small, polycrystalline grains exhibiting diverse morphologies like small rods, irregular parallelepipeds, and spheres, with a wide range of particle size distribution. Nanoparticles of different morphologies were well dispersed. The citrate used in the first stage of the composites’ synthesis acted as a dispersant for the commercial ZnO nanoparticles (which can be covered with organic stabilizers). The carboxylate anions attached to the positively charged surface of ZnO, forming a Zn^2+^–citrate complex, keeping the ZnO NPs away from each other [39]. The elongated particles of commercial ZnO had a mean length of 131.6 +/4.4 nm (N = 150) and the spherical ones had a mean diameter of 43.1 +/1.5 nm (N = 250). During citrate functionalization, the ZnO NPs underwent some morphological changes, especially in the case of the aspherical particles—the length of these nanoparticles increased, with the mean length being 199.5 +/8.2 nm (N =150)—while their diameter remained constant. Citrate ions were preferentially adsorbed on the positively charged zinc (001) plane, favoring the selective growth of the wurtzite crystal [40]. At the same time, the spherical NPs slightly decreased in diameter after the citric acid treatment to 41.1 +/1.1 nm (N = 250). Since the SEM analysis technique cannot distinguish between ZnO, Au, and Ag, the changes that appeared after the Ag and Au synthesis were highlighted through the statistical analysis of the nanoparticle’s dimensions (Appendix A). In both nanocomposites, the number of particles that were less than 50 nm in diameter increased compared with those from the ZP1 sample. TEM analysis was performed on Au and Ag nanoparticles obtained in the absence of ZnO to generate additional information about the shape and dimensions of these nanoparticles. Therefore, the AuNPs appeared to be polycrystalline and near-spherical in shape with diameters ranging from 5 to 35 nm (inset of ZP2 SEM image—Figure 2), and the AgNPs were rounder in shape with sizes between 8 and 40 nm (inset of ZP3 SEM image—Figure 2). Also, transmission electron micrographs showed that the gold nanoparticles were less polydisperse than the AgNPs, with most nanoparticles being around 30 nm in diameter. The EDS analysis proved the presence of Au in the ZP2 nanocomposite and of Ag in the ZP3 sample (Figure 2), and the values were similar to those obtained from the ICP-OES analysis: Au content was 1.12 wt.% and Ag was 0.97 wt.%. All the samples contained carbon from citrate groups, the amount being greater in both nanocomposites than in ZP1.

During the synthesis of noble metal nanoparticles, ZnO nanocrystals acted as seeding material for the nucleation of gold or silver nanocrystals [24]. Sodium citrate was used as a reducing agent for HAuCl_4_ (precursor of AuNPs) and AgNO_3_ (precursor of AgNPs), respectively. The reduction reactions can be listed as the following [29]:4HAuCl_4_ + HOC(COONa)(CH_2_COONa)_2_·2H_2_O + 3H_2_O→3HOOC(COONa) + 4Au + 16HCl (2)
AgNO_3_ + HOC(COONa)(CH_2_COONa)_2_·2H_2_O → 3HOOC(COONa) + Ag + NaNO_3_(3)

Also, the citrate groups performed a dispersant role, preventing the agglomeration of the newly synthesized Ag and Au nanoparticles [41].

The nature of the interactions that occurred in the synthesis processes of the nanocomposites and the newly formed bonds were evaluated through FTIR spectroscopy. Figure 3 presents spectra for citrate-functionalized ZnO (ZP1) and the two nanocomposites (ZP2 and ZP3). All the spectra were characterized by absorption bands observed below 600 cm^−1^, which can be attributed to the vibrational mode of the Zn-O bonds in the wurtzite structure of ZnO [42]. 

In the case of ZP1, three absorption peaks are associated with the vibration mode of the Zn-O bonds, suggesting the coexistence of particles with different sizes. The treatment with citrate removes the organic additives from commercial ZnO (proved by the disappearance of certain bands in the FTIR spectrum of the ZP1 sample in comparison to that of commercial ZnO—Appendix A). The peak centered at 439 cm^−1^, corresponding to the E1 (TO) mode of hexagonal ZnO in the ZP1 spectrum, undergoes a slight shift towards lower wavenumbers in ZP2 and ZP3 due to the formation of nanocomposites. However, FTIR spectroscopy cannot detect specific bonds of Au or Ag atoms, suggesting the existence of an electrostatic interaction between the oxide and metallic particles. The broad band centered at about 3440 cm^−1^ corresponds to the stretching of the O-H bonds, indicating the presence of moisture in all samples. Besides these, in the spectral range of 3000–600 cm^−1^, peaks of different intensities associated with the process of synthesis and stabilization of nanoparticles can be observed, but without affecting the crystalline structure of ZnO, existing mainly near the surface of the oxide. The peaks associated with the mode of symmetric and asymmetric vibration of C-H bonds from saturated organic residues can be observed in the spectral range 3000–2800 cm^−1^. The bands in the range 2400–2000 cm^−1^ indicate the presence of adsorbed CO_2_ from the air on oxide nanoparticles. In the case of samples ZP2 and ZP3, the peaks in the range 1600–800 cm^−1^ are associated with the mode of vibration of the bonds from the trisodium citrate used with a triple role in the synthesis process of metallic nanoparticles: reducing agent, size controller, and steric stabilizer [43]. The bands in the 1580–1400 region can be associated with the symmetric and asymmetric vibration mode of the bonds from the COO^-^ group in the citrate that is electrostatically attached to the nanoparticles.

### 2.2. Optical Properties

The optical absorbance spectra (Figure 4a) showed a strong absorption maximum located in the UV region and attributed to a large excitonic binding energy, as expected for ZnO. Also, some modifications in the absorption spectra can be induced by the small increase in the ZnO particle size (seen in SEM images—Figure 3), which led to the scattering of light and the decrease in the crystallite size and lattice strain of wurtzite during nanocomposite synthesis (Table 2). For both nanocomposites, the absorption in the visible range increased, being more pronounced for ZnO/Au nanocomposites. The increase in optical absorption in the visible range compared to ZP1 may have occurred due to the surface plasmon resonance absorption of silver nanoparticles (between 400–500 nm) and gold nanoparticles (between 500–700 nm), which is evidence for the existence of ZnO–Au and ZnO–Ag interfaces. The stronger shift in the ZnO band position for ZnO/Au can be attributed to the higher electronegativity of gold when compared to silver, meaning that it is more efficient in pulling the electron density towards itself [44].

The band gap energies of the synthesized samples were determined by diffuse reflectance spectra (Figure 4b) [45], applying Equation (1). The results showed a small yet detectable decrease in the energy band of the nanocomposites, from 3.260 eV for ZP1 to 3.255 eV for ZnO/Au and 3.252 eV for ZnO/Ag. This was due to the AgNP and AuNP electrostatic interactions with the ZnO NPs, the decrease in the density of states, and screening of the Coulomb interaction with a smaller surface-to-volume ratio. The decrease in the band gap energy is in good agreement with the redshift observed in the absorbance spectra.

### 2.3. Antimicrobial Activity

The results of the disk diffusion test for antimicrobial susceptibility are presented in Figure 5. As can be observed, ZP2 and ZP3 had a larger zone of bacterial growth inhibition compared to ZP1, for both the Gram-positive and Gram-negative bacteria tested. The antimicrobial effect of the nanomaterials showed a marked dependence on the chemical component of the material introduced into each of the agar discs. The sizes of the inhibition zones (IZ) of all the tested nanomaterials against *P. aeruginosa* were larger than those of IZ against *S. aureus* (Table 3), suggesting that *P. aeruginosa* was more sensitive to the nanomaterial action. The inhibition zones in the case of *P. aeruginosa* had no clear or regular boundaries, especially in the case of the nanocomposites, indicating that ions from the nanomaterials leached out from the disks in a different way compared to the case of the *S. aureus* bacteria dishes. In the *S. aureus* plates, all the inhibition zones were clear, meaning that a gradient of ion concentration was created around all the disks, suggesting that these bacteria were highly sensitive (susceptible) to the antimicrobial agent [46,47]. Also, all the tested nanomaterials prevented bacteria from growing or replicating, being able to kill bacteria.

In the case of the *P. aeruginosa* strain, discs impregnated with ZP1 had a diameter of IZ of 10 mm, the IZ diameter for the ZP2-impregnated disks was almost 80% greater than that observed for the ZP1 impregnated disks, and the IZ diameter increased by 30% for the ZP3 samples. Similarly, in the case of the *S. aureus* strain tests, the ZP1 induced an inhibition zone of 9 mm, while for ZP2 this increased by 55% and for ZP3 by 33%.

The results from the MIC and MBC tests, representing the antimicrobial activity of the nanomaterials dispersed in batch cultures, are summarized in Table 4. By measuring the effect of decreasing concentrations of nanomaterials on *S. aureus* bacteria growth over 24 h, the MICs were found to be 6.25 µg/mL (ZP1), 1.5 µg/mL (ZP2), and 3.25 µg/mL (ZP3), which are lower concentration values compared to those of the same nanoparticles against *P. aeruginosa* (12.5 µg/mL, 3.25 µg/mL, and 6.25 µg/mL, respectively). Similarly, the MBC for nanomaterials against *S. aureus* was found to be 6.25 µg/mL (ZP1), 3.75 µg/mL (ZP2), and 6.25 µg/mL (ZP3), being lower in each case compared to the registered results against *P. aeruginosa* bacteria (25 µg/mL (ZP1), 6.25 µg/mL (ZP2), and 12.5 µg/mL (ZP3)). The results indicated that *S. aureus* (Gram-positive) planktonic cells were more susceptible to citrate-functionalized ZnO, ZnO/Au, and ZnO/Ag than *P. aeruginosa* (Gram-negative). The difference in susceptibility may be attributed to variations in the cell surface characteristics between *P. aeruginosa* and *S. aureus* bacteria, in particular the bacterial cell membrane [48]. Gram-negative bacteria have a periplasmic space that may act as a barrier for the nanoparticles. Also, Gram-negative bacteria produce more proteins that bind to the surface of ZnO NPs, making the interaction between the NPs and bacteria more difficult. Furthermore, Gram-negative bacteria exhibit an overexpression of efflux pumps and porins, which may limit the penetration of nanoparticles into the cell. In the case of *P. aeruginosa*, it has been observed that the bacterium produces a pigment called pyocyanin, which serves as a defense mechanism against nanoparticles [49,50]. Pyocyanin can interact with the ions produced or released by ZnO NPs or AgNPs, thereby neutralizing their antimicrobial effects. These factors contribute to the higher susceptibility of Gram-positive *S. aureus* cells to ZnO NPs compared to Gram-negative *P. aeruginosa* cells.

The proposed mechanism of antimicrobial action is complex, including the release of cations and direct interaction with the cell wall through electrostatic interactions, synergistically with the oxidative damage caused by ROS generation. The release of cations from the nanoparticles is thought to be one of the mechanisms of antimicrobial action in the absence of light. The interactions between cations and the bacterial membrane are facilitated by the negatively charged molecules composing both Gram-positive and Gram-negative cell walls. As a result, NPs are electrostatically attracted to the surface of bacteria, disrupting their cell walls and increasing their permeability. The dissimilarity in antimicrobial activity of nanocomposites may be due to the distinct properties of silver and gold; silver has a higher reactivity, while gold is known for its chemical stability and corrosion resistance [49]. Nanoparticles that came into contact with microbes interact with the bacterial cell membrane and start to intervene in the basic processes of the cells. Among the results of nanoparticle actions, the most important are the damage of the bacterial cell membrane, cellular fluid leakage, DNA and protein disruption, and enzyme deactivation [51]. This interaction with bacteria is dependent on the chemical nature of the nanoparticles. Silver and gold have a strong affinity for different chemical groups from the cell, particularly to SH groups. Ag ions have been proven to block the respiratory chain, inhibiting respiratory enzymes [21], and to disturb the replication of DNA, without inducing resistance to silver ions, although some studies have shown a few exceptions [52]. Compared to silver and ZnO, AuNPs have not been as extensively explored for their antibacterial properties, but now have become a hot research topic [53]. Small gold nanoparticles can perforate the bacterial cell membrane, which results in cell death [54], or can cause depolarization of the membrane potential and DNA damage, causing apoptotic-like death [55]. From the literature, ZnO-based nanoparticles, doped or in conjugation with noble metals, have demonstrated effective antibacterial activity through the formation of ROS (especially OH and singlet oxygen radicals [56]) and the releasing of Zn^2+^ [57], responsible for the cellular damage. Furthermore, the citrate groups present in each synthesized nanomaterial contribute to the overall antimicrobial effect [24].

### 2.4. Photocatalytic Activity

The photocatalytic activities of the citrate–ZnO, ZnO/Au, and ZnO/Ag nanocomposites were assessed by the degradation of MB under UV irradiation. Figure 6a shows the absorption spectra of the aqueous solution of MB with 40 mg ZnO, ZnO/Au, and ZnO/Ag nano-photocatalysts. In the absence of ZP samples, the concentration of MB decreased by only 1.14% under UV irradiation (Figure 6b). When using ZP1 as a photocatalyst, the concentration of MB decreased over the irradiation time, which can be attributed to nanometric dimensions. In the case of the nanocomposite photocatalysts, the MB degradation capability was increased in comparison with that caused by bare ZnO NPs, proving that the presence of AuNPs or AgNPs had a positive effect on the photocatalytic performance. According to the experimental results, the calculated photocatalytic degradation efficiencies were 94.85% (ZP2) and 97.80% (ZP3), while ZP1 achieved 83.12% (Figure 6b). The degradation rate of MB was higher in the case of the nanocomposites compared to ZP1, proven by the higher slope of the ZP2 and ZP3 curves in Figure 6b. The increased photocatalytic effectiveness of the ZnO/Au and ZnO/Ag nanocomposites can be attributed to the synergetic effect of the nanocomposite’s constituents and the specific charge transfer upon constituent contact. The strong electronic interaction between AuNPs and ZnO, and between AgNPs and ZnO, respectively, facilitated the charge transfer from ZnO to the noble metal nanoparticles, leading to better charge separation of the photogenerated electron–hole pairs. The presence of noble metals in ZnO-based nanocomposites can result in a decrease in work functions, leading to an increase in the electron transfer rate and a faster rate of dye degradation [32]. In photocatalysis, the charge carriers that have escaped annihilation migrate to the surface of the catalyst and initiate reactions with the surface-adsorbed species. The holes react with H_2_O molecules, producing hydroxyl radicals, whereas the electrons react with dissolved oxygen, resulting in superoxide radicals or hydroperoxide radicals. All these species contribute to the degradation of methylene blue dye [58].

The photocatalytic efficiency of ZnO/Ag being a little higher than that of ZnO/Ag was in concordance with the values of the band gaps (Figure 4b). Other authors obtained a better improvement in the removal of Congo red dye under UV light irradiation when AuNPs were deposited onto ZnO compared to AgNP deposition [29]. Fageria et al. [44] found that Au-decorated ZnO exhibited better photocatalytic efficiency in comparison with ZnO/Ag, this result being on account of the greater work function value of gold compared to that of silver.

## 3. Materials and Methods

### 3.1. Reagents

ZnO nanoparticles (<100 nm) were purchased from Merck (Darmstadt, Germany). Au (III) chloride trihydrate HAuCl_4_∙3H_2_O (99%), silver nitrate AgNO_3_ (99.999%), sodium citrate C_6_H_5_Na_3_O_7_ (99.0%), and Mueller–Hinton Agar were purchased from Sigma-Aldrich (Darmstadt, Germany) and used without further purification.

### 3.2. Preparation of ZnO/Au and ZnO/Ag Nanocomposites

During nanocomposite synthesis, commercial ZnO NPs were firstly functionalized with citrate, and then nanoparticles of Au or Ag were synthesized in the presence of the citrate-functionalized ZnO NPs. Then, 500 mg of commercial ZnO nanoparticles were dispersed in 100 mL of deionized water at room temperature using an ultrasonic bath for 30 min. In this dispersion, sodium citrate powder (0.7 mM) was added and maintained for 24 h under magnetic stirring. Precursor powder of HAuCl_4_ or AgNO_3_ was added into the mixture followed by magnetic stirring for 2 h in the dark. The resulting precipitate was centrifuged at 10,000 rpm, washed with deionized water 4 times, and dried in an oven at 70 °C. The obtained ZnO/Au powder had a faint purple color while ZnO/Ag was light yellow. For comparison, ZnO NPs functionalized with citrate were washed and dried. The samples were coded according to Table 1.

### 3.3. Characterization Techniques

X-ray diffraction investigations were performed using a 9 kW Rigaku SmartLab diffractometer ($\lambda_{\mathrm{CuK}\alpha 1}$ = 0.154 nm) in parallel-beam configuration. Powder X-ray diffraction (PXRD) investigations were performed in θ/2θ mode. The morphological features of samples were studied using a Quanta Inspect F scanning electron microscope (FEI Company), and their chemical composition by energy dispersive X-ray spectroscopy (EDX). A Fourier Bruker Optics Tensor 27 spectrometer was used to perform transform infrared (FTIR) spectroscopy to study the chemical bond configuration by averaging 64 scans with a resolution of 4 cm^−1^, at room temperature, using the potassium bromide pellet method. The sizing of the ZnO nanoparticles and synthesized composite powders was performed using ImageJ software1.8.0. The amounts of Au and Ag in the synthesized nanocomposites were determined by inductively coupled plasma optical emission spectrometry (ICP-OES) using an Optima 5300 DV (Perkin ElmerInc., Waltham, MA, USA). The optical absorbance spectra of the powders were recorded at room temperature using a Cary 5000 spectrophotometer (Agilent Technology, Santa Clara, CA, USA). The diffuse reflectance spectra were recorded from 200 to 700 nm in 1 nm intervals. All spectra were baseline-corrected with a 100% R baseline collected over the same spectral domain using a white polytetrafluoroethylene standard sample. The band gap energy, *Eg*, was estimated from the Kubelka–Munk function F(R) defined by Equation (4):K/S = (1 − R)^2^/2R(4)
where K and S are the K-M absorption and scattering coefficients, and R is the reflectance. The K-M function was employed to determine the band gap energy (Eg) of the ZnO powders by replacing the optical absorption coefficient α in the analogous Tauc plots:αhν = A (hν − E_g_)^n^(5)
where A is a constant, hν is the incident photon energy, and the exponent n = ½ and 2 for direct allowed transition and indirect allowed transitions, respectively.

### 3.4. Antibacterial Testing

#### 3.4.1. Antibacterial Susceptibility

Antibacterial susceptibility testing was determined using the Kirby–Bauer disk diffusion susceptibility test in the dark on a standardized Mueller–Hinton culture medium. *S. aureus (ATCC25923)* and *P. aeruginosa (ATCC27853*) were used as representative strains of Gram-negative and Gram-positive bacteria, according to the Clinical and Laboratory Standards Institute (CLSI) and the European Committee on Antimicrobial Susceptibility Testing (EUCAST) [46,47]. The bacterial inoculum was prepared by suspending 3–5 colonies in physiological serum. Turbidity of 0.5 on the McFarland scale (0.5 McFarland = 1.5 × 10^8^ cells per mL) was measured by a nephelometer. Sterile paper discs (6 mm in diameter) were impregnated with 30 μL of an antimicrobial agent (30 mg/mL) and placed on an agar plate with bacterial inoculum. After 15 min, the plates were placed in the thermostat at 35 °C in aerobiosis, for 24 h. The sizes of the bacterial growth inhibition zones were measured in millimeters (mm). Each test was performed in triplicate and the average of the values was calculated.

#### 3.4.2. Minimum Inhibitory Concentration (MIC) and Minimum Bactericidal Concentration (MBC) Assays

The MIC determination was performed by a microdilution method according to the protocol M7-A10 of the Clinical and Laboratory Standards Institute (CLSI) [31] using a standardized Mueller–Hinton medium. Dispersions of 100 µg/mL of all nanomaterials in 4% dimethyl sulfoxide (DMSO) were added to micro-wells in a serial dilution, giving concentrations of 50, 25, 12.5, 6.25, 3.75, and 1.5 µg/mL.

The bacterial suspension containing approximately 5 × 10^5^ colony-forming units/mL was prepared from a 24 h culture plate. From this suspension, 100 μL was inoculated into each well. To determine the sensitivity of the microorganisms, positive control experiments were conducted for bacterial strains in standardized Mueller–Hinton medium and DMSO, while for sterility only control broth and DMSO. The microplates were incubated at 37 °C for 24 h, in darkness. Afterward, the concentrations at which the tested nanomaterials inhibited the growth of (MIC) or completely killed (MBC) *S. aureus* (ATCC25923) and *P. aeruginosa* (ATCC27853) were established. All experiments were performed in triplicate and the results represent the average of the three measurements.

### 3.5. Photocatalytic Testing

The assessment of photocatalytic activity was performed by the degradation of methylene blue (MB) dye in the presence of the synthesized samples, under UV irradiation (254 nm) from a mercury UV lamp at 100 KW. First, 40 mg of each photocatalyst was dispersed in a 40 mL aqueous solution of MB (0.5 mg/mL). Before UV exposure, the suspension was stirred (at 800 rpm) for 30 min in the dark to reach an adsorption–desorption equilibrium. The temperature of the photo-oxidation reaction was maintained at 25 °C. At a regular irradiation time interval, the MB content was quantified by measuring the absorbance using a UV–Vis spectroscopy and a microplate reader with an Infinite 200 PRO NanoQuant fluorescence spectrometer (Tecan, Switzerland). The efficiency of dye degradation was calculated using the relation:(6)$$\eta =\frac{A_{0}-A_{t}}{A_{0}}\;\cdot \;100$$
where $A_{0}$ is the initial absorbance and $A_{t}$ is the at time ‘*t*’.

### 3.6. Statistical Analysis

The experimental results were subjected to statistical analysis using the one-way ANOVA (analysis of variance) test, and a significance level of *p* ≤ 0.05 was applied to determine whether the results were statistically significant.

## 4. Conclusions

ZnO/Au and ZnO/Ag nanocomposites were successfully synthesized by a simple chemical approach through the citrate functionalization of commercial ZnO NPs, followed by the synthesis of Au or Ag nanoparticles (1 wt.%) on citrate–ZnO NPs through the citrate reduction method. XRD analysis revealed a relaxation of the wurtzite lattice simultaneously with the formation of dislocations within it. The citrate–ZnO, ZnO/Au, and ZnO/Ag morphologies were investigated and showed polycrystalline grains with different morphologies: small rods, irregular parallelepipeds, and spheres. The formation of gold and silver nanoparticles between ZnO NPs was proven by EDS and ICP-OES and indicated through the modification of the nanoparticle size distribution in SEM images, after noble metal synthesis. The citrate acted as a reducing agent, size controller, and steric stabilizer. These ions adsorbed on the ZnO surface, as well as on AuNPs and AgNPs, as exposed through specific bands in FTIR spectroscopy. The presence of 1% noble metal in the ZnO matrix expanded the spectral range of ZnO into the visible region, which is useful in the production of more ROS. Disk diffusion tests showed a greater biocidal potential of the nanomaterials against *P. aeruginosa* than in the case of *S. aureus*, and the bacteria were the most sensitive to ZnO/Au. The same nanocomposite also yielded the best results against *S. aureus*, with an inhibition of bacterial growth at MIC 1.5 µg/mL and a minimum bactericidal effect at MBC 3.75 µg/mL. Overall, the nanocomposites proved to be better antimicrobial agents than ZnO alone due to a positive synergistic effect between the noble metals and ZnO antimicrobial against *P. aeruginosa* and *S. aureus* bacterial strains.

The photocatalytic performance of the ZnO/Au and ZnO/Ag nanocomposites was tested in the degradation of MB dye under UV irradiation and compared with that of citrate–ZnO. The results indicated that both noble metals improved the overall photocatalytic degradation efficiency of MB dye. The maxima of the photocatalytic degradation efficiency were ∼98% (ZP3) ∼and 95% (ZP2), while ZP1 achieved ∼83% after 60 min of UV irradiation. The differences in AuNP and AgNP activity may be related to the differences in their work functions. These findings confirm that Au and Ag nanoparticles in contact with nano-ZnO can delay the recombination of the photogenerated electron–hole pairs. ZnO/noble metal composites prepared through the chemical route proved to be suitable and effective for antimicrobial and photocatalytic applications.

## Figures and Tables

**Figure 1 ijms-24-16939-f001:**
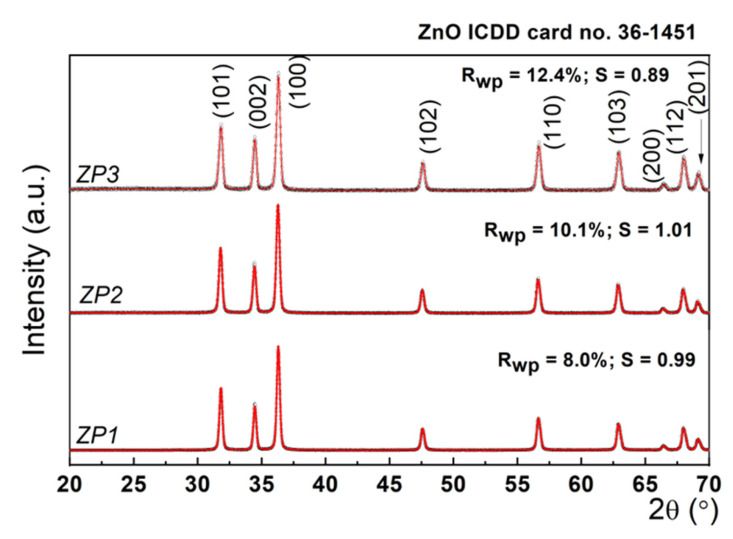
Experimental XRD patterns (black points) and simulated data in the framework of the Rietveld refinement (red line).

**Figure 2 ijms-24-16939-f002:**
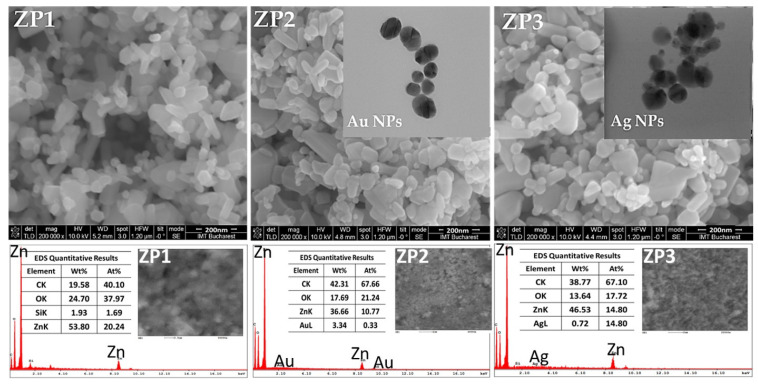
SEM micrographs, EDS elemental analysis of ZP1, ZP2, and ZP3. The insets are TEM micrographs of AuNPs and AgNPs.

**Figure 3 ijms-24-16939-f003:**
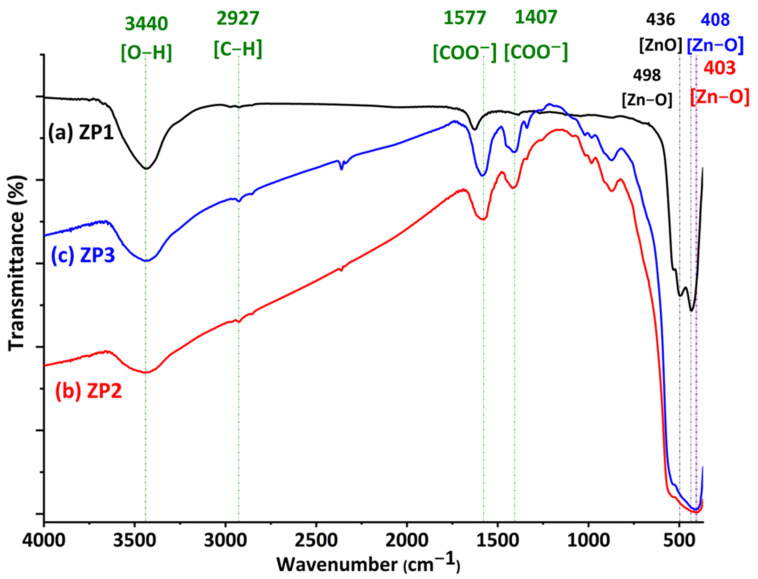
FTIR spectra for (a) ZP1, (b) ZP2, and (c) ZP3 samples.

**Figure 4 ijms-24-16939-f004:**
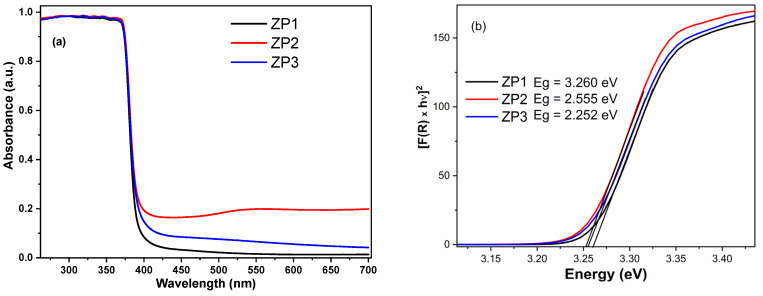
(**a**) UV–Vis optical absorbance and (**b**) Kubelka–Munk transformation for determination of direct energy band gap of ZP1, ZP2, and ZP3 samples.

**Figure 5 ijms-24-16939-f005:**
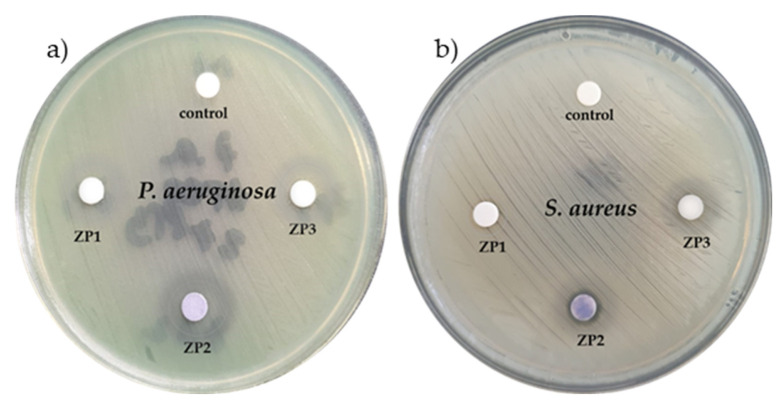
The zones of inhibition on MH agar plate of (**a**) *P. aeruginosa* and (**b**) *S. aureus*.

**Figure 6 ijms-24-16939-f006:**
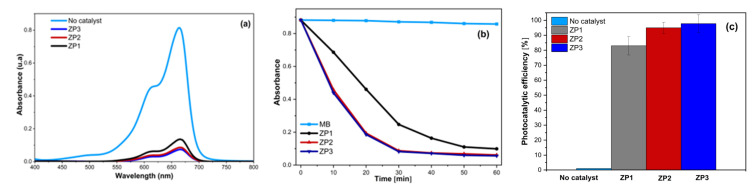
(**a**) The absorption spectra of MB solutions in the presence of ZP1, ZP2, and ZP3 samples (40 mg ZP/20 mg MB after 1h of UV irradiation), (**b**) change in MB concentration with irradiation time, and (**c**) photocatalytic efficiency of ZP1, ZP2, and ZP3 samples against MB (40 mg ZP/20 mg MB 1 h of UV irradiation).

**Table 1 ijms-24-16939-t001:** The codes of the nanomaterials.

Sample Code	Nanomaterials	
ZP1	citrate-functionalized ZnO	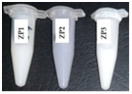
ZP2	ZnO/Au nanocomposite	
ZP3	ZnO/Ag nanocomposite	

**Table 2 ijms-24-16939-t002:** Unit cell parameters, mean crystallite size, lattice strain, and fitting parameters.

Sample	Unit Cell Parameters (nm)		Crystallite Size (nm)	R_wp_ %	S	Lattice Strain (%)
	a	c				
ZP1	0.325	0.52	31.1	8.00	0.9933	0.73
ZP2	0.325	0.52	27.5	10.12	0.9110	0.64
ZP3	0.325	0.52	25.3	12.40	0.8935	0.42

**Table 3 ijms-24-16939-t003:** Mean values of IZ (in mm) produced by ZP1, ZP2, and ZP3 on the tested bacteria.

Sample	*P. aeruginosa*	*S. aureus*	*p*-Value
ZP1	10 ± 1.25	9 ± 0.81	<0.05
ZP2	18 ± 1.41	14 ± 0.47	
ZP3	13 ± 0.94	12 ± 0.81	

**Table 4 ijms-24-16939-t004:** MIC (µg/mL) and MBC (µg/mL) for ZP1, ZP2, and ZP3 on *P. aeruginosa* and *S. aureus*.

Sample	*P. aeruginosa*		*S. aureus*	
	MIC	MBC	MIC	MBC
ZP1	12.5	25	6.25	6.25
ZP2	3.75	6.25	1.5	3.75
ZP3	6.25	12.5	3.75	6.25

## Data Availability

Data is contained within the article and Appendix A.

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
