# Peer review of "Antibacterial and Photocatalytic Activity of ZnO/Au and ZnO/Ag Nanocomposites"

_ijms, 2023, doi:10.3390/ijms242316939_

Round 1

Reviewer 1 Report

Comments and Suggestions for Authors

In the work the antibacterial and photocatalytic properties of nanocomposites based on ZnO, Ag and Au are considered. The work is not original, since nanocomposites of this type and their relative capabilities are well known and characterized in the literature. Furthermore, I detected serious lacks in the characterization of materials, in particular:

1)     The presence of Ag and Au in the nanocomposite is not clearly demonstrated:

a.      The typical signals of these metals are absent from the XRD spectra (Fig.1) an this is at least unusual if they are instead detected by EDS, as the authors claim.

b.      Itensity and distribution in the EDS maps are not unequivocal, especially for Au, as no morphological characteristic is recognizable that could confirm the attribution of the bright dots to some particle.

c.      In the text ICP-OES measurements are mentioned which would confirm the presence of metals, but there is no other trace of the technique which is not even mentioned in the materials and methods.

d.      The dimensional test proposed by the authors is not acceptable, as the comparison is carried out considering two different techniques (SEM vs TEM) and above all by referring to particles synthesized in non-comparable conditions (with and without ZnO NPs)

2)     The decrease in size of ZnO NPs following citrate functionalization is unexpected; what is the possible explanation? Greater characterization is fundamental given that this step influences the subsequent synthesis of metal particles.

3)     Assuming that Au and Ag are actually present in the samples, it is unclear what the structure of the nanocomposite is. Are core-shell structures formed or should NPs be considered standalone?

4)     Regarding FTIR spectra, in the text the reference spectrum is assigned to commercial ZnO, but ZP1 corresponds to the citrate functionalization step in Materials and Methods. The untreated spectrum should be included to demonstrate that the replacement of the organic stabilizers with citrate was efficient.

5)     Why are the signals between 1600-800 cm-1 in ZP2 and ZP3 associated with trisodium citrate not also visible in the spectrum of ZP1? Are they due to the presence of free citrate, despite the 4 centrifugation and washing procedures? Could it possibly have an effect on the properties of the particles?

Moreover, the zones of bacterial growth inhibition are not as evident in the agar plates shown in figure 5, indicating limited activity of the samples. The distinction between bacteriostatic and bactericidal activity based on the appearance of the inhibition zones requires literature references.

Some minor corrections, for completeness:

a)      Equation 4 should be equation 1, being the first equation introduced in the text. The reduction reactions for the syntesis of Au and Ag NPs deserves a numeration too.

b)     Table 2: the correspondence between the acronym and the sample type should be explicitly clarified. The same applies for every acronym in the text.

It seems to me that the work was adapted from another template without the foresight of adapting the text to the new position of the materials and methods section, which is a sign of little care.

c)      The scales in figure 2 are illegible. Each figure on the panel should be marked with a letter, so that it is easy to refer to it in the text and caption. The latter presents an inaccuracy in the sample labels. The central part of the figure showing the EDS spectra is a low-resolution screenshot of which neither the scale nor the labels are clearly distinguishable. Unnecessary character strings should be eliminated.

Comments on the Quality of English Language

The English language is fine. I detected a few typos in the text which could be easily corrected.

Reviewer 2 Report

Comments and Suggestions for Authors

The manuscript has strong potential and is generally well written.  There are a number of questions that should be addressed regarding the antimicrobial activity:

1. The authors MUST put table 1 near the beginning of the article.  There are no references to the compositions of ZP1-3 until section 4, which made initial reading very confusing.

2. The authors note a diffuse ring of bacterial growth in the disc-diffusion assays.  Are these viable bacteria in the ring or bacteria which have died as result of exposure?  This is important in the interpretation of results.

3. Considering the photoactivity, were the antimicrobial tests performed in light or dark?  Either way, why was only one condition tested?

4.  The authors must elucidate on the MIC methods, specifically volumes of sample and bacterial solutions used.  DMSO is known to have antimicrobial activity at high concentrations, so explicitly defining the concentration, and showing control experiments with DMSO-vehicle alone, are critical for proper interpretation of the MIC results (https://doi.org/10.1002/jps.2600580708).

5. Does the citrate modification of the particles impact pH of the media for bacterial growth?

6. Overall, i am curious about the authors interpretation of the structure/activity of these composites.  A clear paragraph regarding the proposed model (i.e. Ag ions combining with Zn nanoparticles..., or whatever the model would be).  The authors talk about multiple possible mechanisms and models, but it would be good to have an interpretation based on this data.

7.  Controls for MIC would be useful with DMSO, and simple solutions of Zn, Ag, and Au, to compare results.  Can the effects be attributed to one specific species?  

Comments on the Quality of English Language

Needs a proofreading, but overall in OK shape.

Reviewer 3 Report

Comments and Suggestions for Authors

The paper entitled “Antibacterial and photocatalytic activity of ZnO/Au and ZnO/Ag nanocompositesis globally comprehensive.

Suggestions are described below:

1)    Abstract should be quantitative as possible for rapid comparison with similar studies. Avoid imprecise terms such as “enhanced …..but how much? 20% two-fold…more effective? But how much more, 10-fold? Abs should be rewritten. Do not jump to conclusions before described the results.

2)    Introduction: at the end of the intro, it is also not clear what is the main message and relevant points and highlights of the paper that should be emphasize at this stage. Specific objectives should be referred highlighting what is new and timely within this topic by 2023.

3)    The results are not globally well described. For instance see section 2.3. Avoid use imprecise terms such as significantly larger.

4)    Was the data validated by statistical analysis? If so, there is no need to refer that the values were “significantly”. However, the quantitative information about the decrease (or increase) is fundamental. The data should be described first. The reader should visualize the data after a correct description.

5)    At line 123 is written: “The results showed a significant difference in the antibacterial activity of ZnO/Au and ZnO/Ag nanocomposites”. Please quantify this difference. How much? From x to y? 10 fold?

6)    Globally conclusions should followed the order of presentation of the paper with partial conclusions first and then global conclusions. Quantitative information regarding the observed effects is also welcome, avoiding imprecise terms.

Round 2

Reviewer 2 Report

Comments and Suggestions for Authors

The authors have improved the manuscript.  However, there are two remaining issues that were not fully addressed from the comments/responses.

1. The authors must include statements that MIC and zone-of-inhibition studies were done in the dark, especially in the methods section.  This is a critical component.  While it does not appear the authors are able to repeat these in a "light" setting, this would have made a much more convincing result.

2. The authors misunderstood the comment about the diffuse zones of growth in the zone-of-inhibition studies.  The visual depictions are clear.  However, what is actually HAPPENING in those areas?  Are these non-confluent bacteria that were killed in the process of growth?  or are these simply lower densities of bacteria in those areas because growth is inhibited or slowed by the release of the ions?  This can be very simply assayed by spot-culturing those areas after imaging and subculturing on fresh media (liquid or solid) without antimicrobial.  Regardless, the authors should comment on these diffuse zones.
